# The Role of Vitamin D in SARS-CoV-2 Infection and Acute Kidney Injury

**DOI:** 10.3390/ijms23137368

**Published:** 2022-07-01

**Authors:** Ming-Chun Hsieh, Po-Jen Hsiao, Min-Tser Liao, Yi-Chou Hou, Ya-Chieh Chang, Wen-Fang Chiang, Kun-Lin Wu, Jenq-Shyong Chan, Kuo-Cheng Lu

**Affiliations:** 1Department of Internal Medicine, Taoyuan Armed Forces General Hospital, Taoyuan 235, Taiwan; b45170@gmail.com; 2Division of Nephrology, Department of Internal Medicine, Taoyuan Armed Forces General Hospital, Taoyuan 235, Taiwan; ajie1124@gmail.com (Y.-C.C.); wfc96076@yahoo.com.tw (W.-F.C.); ndmc6217316@yahoo.com.tw (K.-L.W.); jschan0908@yahoo.com.tw (J.-S.C.); 3Division of Nephrology, Department of Internal Medicine, Tri-Service General Hospital, National Defense Medical Center, Taipei 114, Taiwan; 4Department of Life Sciences, National Central University, Taoyuan 320, Taiwan; 5School of Medicine, Fu-Jen Catholic University, New Taipei City 242, Taiwan; liaoped804h@yahoo.com.tw; 6Institute of Molecular and Cellular Biology, National Tsing Hua University, Hsinchu 300, Taiwan; 7Department of Pediatrics, Taoyuan Armed Forces General Hospital, Taoyuan 325, Taiwan; 8Department of Pediatrics, Tri-Service General Hospital, National Defense Medical Center, Taipei 114, Taiwan; 9Department of Medicine, Cardinal Tien Hospital, School of Medicine, Fu Jen Catholic University, New Taipei City 242, Taiwan; liaoped804h@aftygh.gov.tw; 10Division of Nephrology, Department of Medicine, Taipei Tzu Chi Hospital, Buddhist Tzu Chi Medical Foundation, New Taipei City 231, Taiwan; 11Division of Nephrology, Department of Medicine, Fu-Jen Catholic University Hospital, School of Medicine, Fu-Jen Catholic University, New Taipei City 242, Taiwan

**Keywords:** vitamin D deficiency, antioxidant, anti-inflammatory effects, acute kidney injury, severe acute respiratory syndrome coronavirus 2 (SARS-CoV-2), coronavirus disease 2019 (COVID-19)

## Abstract

Vitamin D has been described as an essential nutrient and hormone, which can cause nuclear, non-genomic, and mitochondrial effects. Vitamin D not only controls the transcription of thousands of genes, directly or indirectly through the modulation of calcium fluxes, but it also influences the cell metabolism and maintenance specific nuclear programs. Given its broad spectrum of activity and multiple molecular targets, a deficiency of vitamin D can be involved in many pathologies. Vitamin D deficiency also influences mortality and multiple outcomes in chronic kidney disease (CKD). Active and native vitamin D serum levels are also decreased in critically ill patients and are associated with acute kidney injury (AKI) and in-hospital mortality. In addition to regulating calcium and phosphate homeostasis, vitamin D-related mechanisms regulate adaptive and innate immunity. Severe acute respiratory syndrome coronavirus 2 (SARS-CoV-2) infections have a role in excessive proinflammatory cell recruitment and cytokine release, which contribute to alveolar and full-body endothelial damage. AKI is one of the most common extrapulmonary manifestations of severe coronavirus disease 2019 (COVID-19). There are also some correlations between the vitamin D level and COVID-19 severity via several pathways. Proper vitamin D supplementation may be an attractive therapeutic strategy for AKI and has the benefits of low cost and low risk of toxicity and side effects.

## 1. Introduction

Vitamin D is obtained from fortified foods, dietary supplements, and exposure to sunlight. Vitamin D from the diet and skin is transported in the blood by circulating vitamin D-binding protein (DBP) to the liver. In the liver, vitamin D is metabolized by vitamin D-25-hydroxylase to 25-hydroxyvitamin D, the major circulating metabolite used to determine a patient’s vitamin D status [1]. Almost all 25-hydroxyvitamin D bound to circulating DBP is filtered by the kidneys and reabsorbed by the proximal convoluted tubules. In the proximal renal tubules, 25-hydroxyvitamin D is hydroxylated by the enzyme 25-hydroxyvitamin D3 1α-hydroxylase (CYP27B1) to its active form, 1,25-hydroxyvitamin D [2]. The production of 1,25-hydroxyvitamin D is regulated by serum calcium and phosphorus, plasma fibroblast growth factor 23 (FGF23), and parathyroid hormone levels [3,4]. Appropriate vitamin D supplements can prevent some chronic diseases, such as diabetes mellitus, cardiovascular disease, and chronic kidney disease (CKD), by regulation of oxidative stress through the following ways: inducing the expression of several molecules involved in the antioxidant defense system including glutathione, glutathione peroxidase, superoxide dismutase (SOD); suppressing the expression of nicotinamide adenine dinucleotide phosphate hydrogen (NADPH) oxidase [1,2,3,4,5].

In addition, the vitamin D receptor (VDR) is widely distributed in more than 38 types of tissues [5]. Vitamin D also plays nonclassical roles in cell differentiation and proliferation and has an immunomodulatory effect [6]. This immunomodulatory effect was based on the widely expressed VDR, which is present in T and B lymphocytes, macrophages, and antigen-presenting cells [7,8]. Vit-D induces tolerogenic dendritic cells (DCs) and increases the expression of immunoglobulin-like transcript 3 (ILT3), an important regulator of dendritic cell tolerance, resulting in increased numbers of CD4+CD25+ regulatory T cells [8]. In addition, 1,25-dihydroxyvitamin D increased insulin production, myocardial contractility, the reproductive system, and hair growth and inhibited renin synthesis. Vitamin D may play an important role in modifying the risk of cardiometabolic outcomes, including hypertension, cardiovascular diseases, and type 2 diabetes mellitus [9,10,11,12].

Acute kidney injury (AKI) is a syndrome of different etiologies that is characterized by a rapid decline in glomerular filtration. In 2002, the Acute Dialysis Quality Initiative group proposed the RIFLE classification that defined the three grades of increasing severity (i.e., risk of acute renal failure; injury to the kidney; failure of kidney function) and two outcome classes (i.e., loss of kidney function and end-stage kidney disease) [13]. In 2005, the Acute Kidney Injury Network (AKIN) group modified the AKI definition. This new staging system classified patients with a change in serum creatinine (sCr) concentration ≥0.3 mg/dL (≥26.4 μmol/L) within 48 h as having AKIN stage 1, whereas patients receiving renal replacement therapy (RRT) were included in AKIN stage 3. RIFLE-Risk was classified as Stage 1; RIFLE-Injury and Failure were classified as Stages 2 and 3, respectively; the two outcome classes of RIFLE-Loss and RIFLE-End-Stage Kidney Disease were removed [14,15]. AKI is one of the major causes of morbidity and mortality in hospitalized patients, especially in intensive care centers. Progressive AKI leads to the depletion of renal function, which causes retention and accumulation of phosphate. Phosphate acts as a downregulator of 1-hydroxylase, an enzyme involved in 1,25(OH)2D synthesis and, therefore, decreases vitamin D production. In addition, the progressive loss of active nephrons contributes to attenuating vitamin D synthesis [16]. The incidence of vitamin D insufficiency in critically ill patients has been reported to range from 26% to 82% [17,18]. Two large observational cohort studies showed that vitamin D deficiency (serum 25(OH)D < 15 ng/mL) prior to hospital admission or at the time of critical care is independently associated with increased morbidity and mortality [19,20]. This insufficiency may worsen existing immune and metabolic dysfunctions in critically ill patients, leading to worse outcomes [21]. Both AKI and vitamin D deficiency are common in critically ill patients, and both are associated with increased mortality [22].

The benefit of vitamin D supplements in preventing acute respiratory tract infections was observed via a meta-analysis of 11,321 participants and other reviews [23,24,25,26,27]. The possibility of decreasing the risk of respiratory tract infections, including coronavirus disease 2019 (COVID-19), may contribute to several immune pathways, such as stimulating antiviral mechanisms, reducing proinflammatory cytokines, modulating concentrations of ACE2, and decreasing the chances of endothelial dysfunction [28]. COVID-19 not only causes respiratory disease but also induces the dysfunction or failure of multiple organs in severe cases. The kidney is the second most commonly affected organ after the lungs. COVID-19-associated AKI is linked to an increased risk of mortality and comorbidities.

## 2. Antioxidant and Renoprotective Effect of Vitamin D in AKI Animal Models

Many AKI animal models (Table 1) have also shown that vitamin D has a renoprotective effect. In a contrast-induced AKI model, paricalcitol caused a reduction in unfavorable histopathological findings via its antioxidant effects by inhibiting lipid peroxidation [29]. In a gentamicin-induced AKI model, paricalcitol restored impaired renal function by inhibiting renal inflammation and fibrosis via the interruption of the nuclear factor-kappaB (NF-κB)/extracellular signal-regulated kinase (ERK) signaling pathway and preservation of tubular epithelial integrity via the inhibition of the epithelial–mesenchymal transition (EMT) process [30]. Although another study indicated that the progression of gentamicin-induced AKI was not alleviated by vitamin D treatment, it probably has some beneficial effects on the renin–angiotensin system (RAS) by lowering blood pressure and increasing urine volume as well as a promising effect on the antioxidant system [31]. The NF-κB signaling pathway was also found to have a positive correlation with SARS-CoV-2-related AKI [32].

In a cisplatin-induced and cyclosporine-induced AKI model, paricalcitol may ameliorate cisplatin-induced renal injury by suppressing fibrotic, apoptotic, and proliferative factors via a mechanism that may include the inhibition of transforming growth factor beta-1 (TGF-β1), suppression of mitogen-activated protein kinase signaling (MAPK), and attenuation of p53-induced apoptosis [44,45]. In an ischemia/reperfusion-induced animal AKI model, the renoprotective effect of vitamin D occurred via peroxisome proliferator-activated receptor gamma (PPAR-γ) [46], and pretreatment with paricalcitol also had a renoprotective effect, possibly via Toll-like receptor 4 (TLR4)/NF-κB-mediated inflammation [47]. In an obstructive nephropathy model, paricalcitol preserved tubular epithelial integrity via the suppression of EMT [48,49].

In a lipopolysaccharide (LPS)-induced AKI model, vitamin D_3_ pretreatment had different effects including (1) significantly attenuating LPS-induced renal inflammatory cytokines, chemokines, and adhesion molecules [33] and reinforcing the interaction between renal VDR and the NF-κB p65 subunit; (2) alleviating LPS-induced renal glutathione (GSH) depletion and lipid peroxidation and attenuating serum and renal NO production and protein nitration by regulating oxidant and antioxidant enzyme genes [34]. These results provide a mechanistic explanation for vitamin D3-mediated anti-inflammatory and antioxidative activities.

The vitamin D analogues protect the kidney by targeting three major pathways: the local RAAS, antioxidation, and the NF-κB pathways. In contrast to the recognized importance of vitamin D in CKD patients, the role of vitamin D in AKI patients is not as well defined. It is reasonable to hypothesize that the manner by which vitamin D deficiency may predispose critically ill patients to AKI is related to the innate and adaptive immune response.

## 3. Vitamin D and the Renin–Angiotensin–Aldosterone System (RAAS)

Previous studies have found an inverse correlation between changes in vitamin D and changes in plasma renin activity. Compared with individuals with sufficient 25-hydroxyvitamin D levels (i.e., ≥30.0 ng/mL), those with 25-hydroxyvitamin D deficiency (i.e., <15.0 ng/mL) had higher circulating angiotensin II (Ang II) levels and significantly blunted renal plasma flow responses to infused Ang II. These data suggest that low plasma 25-hydroxyvitamin D levels may result in the upregulation of the RAAS in otherwise healthy humans [35].

Subsequently, mechanistic studies have demonstrated that renin gene expression is increased in the kidneys of VDR-null mice, which was accompanied by increased plasma Ang II levels, hypertension, and cardiac hypertrophy [38]. Conversely, treatment with calcitriol reduced renal renin production independent of calcium and parathyroid hormone (PTH). Calcitriol binds to the VDR and blocks the formation of CRE–CREB–CBP complexes in the promoter region of the renin gene, thus reducing its level of expression [39].

Several experimental studies have confirmed that the renoprotective effects of vitamin D analogue alone can improve proteinuria, glomerulosclerosis, and interstitial infiltration and reduce renal oxidative stress. Combined treatment with paricalcitol and losartan suppressed the induction of fibronectin, TGF-β, and monocyte chemoattractant protein-1 and reversed the decline in the slit diaphragm proteins nephrin, Neph-1, ZO-1, and alpha-actinin-4 [37]. A VDR agonist would provide additional renoprotection via its negative regulation of renin [36]. Paricalcitol has been shown to suppress the expression of renal TGF-β1 and its type 1 receptor, restore VDR abundance, block epithelial to mesenchymal transition, and inhibit cell proliferation and apoptosis [40]. Experiments using VDR-null mice indicated that VDR attenuated renal inflammation at least, in part, by suppressing the RAS [41]. In the VITAL study, the administration of paricalcitol in addition to RAAS blockade further reduced albuminuria compared with RAAS blockade alone in patients with diabetic nephropathy [42].

Although animal and clinical studies have provided important mechanistic clues regarding the crosstalk between RAAS and vitamin D, they are unable to show the translational benefits of vitamin D-mediated RAAS blockade on AKI [43,50].

SARS-CoV-2 binds to the ACE2 receptor expressed on the surface of lung epithelial cells, which causes downregulation of the ACE2 receptor and then leads to excessive presence of Ang II. A high concentration of Ang II may facilitate AKI [28].

## 4. Vitamin D Deficiency and the Risk of AKI

Vitamin D deficiency, which is defined as a serum 25-hydroxyvitamin D level below 50 nmol/L (20 ng/mL), was linked to several types of cancer and autoimmune and metabolic diseases. Vitamin D deficiency is found worldwide [51,52]. In the United States, vitamin D insufficiency (serum 25-hydroxyvitamin D (25(OH)D) < 28 ng/mL) was present in approximately 41% of men and 53% of women [53]. In addition to calcium homeostasis and bone metabolism, vitamin D also plays a role in improving glucose control, thus reducing the need for erythropoiesis-stimulating agents, modulating inflammatory and immune responses, and regulating the RAAS as well as cellular proliferation, differentiation, and apoptosis [50,54].

In animal models of sepsis, the administration of 1,25(OH)D was correlated with improved blood coagulation parameters in sepsis-induced disseminated intravascular coagulation [55]. Another study showed that decreased absolute levels of DBP were consistent in early sepsis and were a prognostic factor for disease severity [56]. 1,25(OH)D can also modulate the levels of inflammatory cytokines and may play a role in LPS-induced immune activation of endothelial cells during Gram-negative bacterial infections.

Jeng et al. [57] found a significantly lower plasma 25(OH)D concentration in patients with sepsis than in healthy controls. Those authors suggested that a low level of circulating 25(OH)D was associated with low cathelicidin. Among mechanically ventilated patients, a 25(OH)D level < 20 ng/mL was associated with a significantly shorter average survival time compared with that of patients with a normal serum level [58]. The 25(OH)D concentration may be either a biomarker of survival or a cofactor in severely ill patients. In critically ill surgical patients, 25(OH)D levels < 20 ng/mL have a significant impact on organ dysfunction, infection rates, and length of stay [58,59]. In some systematic reviews and meta-analyses, vitamin D supplementation also ameliorated ventilator demands, ICU admission and mortality rates in COVID-19 patients [60].

Large retrospective observational cohort studies have demonstrated that vitamin D deficiency, defined as a serum 25(OH)D level < 15 ng/mL, both at preadmission and at the time of critical care, was independently associated with increased morbidity and mortality in intensive care unit (ICU) patients [61]. These data also indicated that preadmission vitamin D deficiency was a significant predictor of AKI. The association between 25(OH)D and AKI was not dependent on the timing of the prehospital 25(OH)D draw. In a secondary analysis, a threshold level of 25(OH)D < 21 ng/mL was significantly associated with AKI with RIFLE-Injury and Failure stage [22]. Another observational cohort study demonstrated the absence of an association between the serum 25(OH)D level at the time of AKI diagnosis and 90 day all-cause mortality in patients with AKI. This result may be because most patients enrolled in the study were too young [62].

In a recent prospective cohort study of 30 individuals with AKI and 30 controls from general hospital wards and ICUs, 25(OH)D levels were inversely correlated with sepsis severity. The principal finding of that study was that the levels of bioavailable 25(OH)D were inversely associated with the severity of sepsis and hospital mortality among patients with AKI. Because the levels of the major metabolite of vitamin D were not elevated in AKI, the reduced levels of 25(OH)D resulted from decreased production and were not related to FGF23. The strong association between the severity of sepsis and bioavailable 25(OH)D vs. total 25(OH)D levels may be related to the selective uptake of bioavailable 25(OH)D by nontraditional target organs including macrophages [63]. Although the exact mechanism underlying this association is unknown, larger studies including serial measurements of 25-hydroxyvitamin D are needed to determine how vitamin D status changes with the progression of AKI and whether vitamin D status at different stages is associated with prognosis [22]. Figure 1 integrates the hypothesis of the association between vitamin D deficiency and AKI.

## 5. Vitamin D and the Immune System

The importance of vitamin D in immune regulation is highlighted by the fact that VDR is expressed in activated inflammatory cells, T-cell proliferation is inhibited by 1,25(OH)2D3, and activated macrophages produce 1,25(OH)2D3 [64]. The innate immune response involves the activation of TLRs on polymorphonuclear cells, monocytes, macrophages, and several epithelial cells. The earliest evidence of the effect of vitamin D on innate immunity came from ultraviolet B (UVB)-irradiated sheep’s wool lanolin, which is a major source of vitamin D [65]. The action of vitamin D on macrophages includes the ability to stimulate the differentiation of precursor monocytes into more mature phagocytic macrophages [66]. Macrophages have their own 1α-hydroxylase and require sufficient ambient levels of 25(OH)D substrate to generate internal 1,25(OH)2D3. Unlike renal 1α-hydroxylase, the 1α-hydroxylase produced by macrophages is not suppressed by elevated calcium or by 1,25(OH)2D3 and is upregulated by immune stimuli, such as interferon gamma (IFN-γ) and lipopolysaccharide (LPS) [67]. In monocytes, the activation of TLR2 induced interleukin-15 (IL-15) secretion and bacterial killing via three key mechanisms: induction of CYPB27B1 (1α-hydroxylase) gene expression, an increase in the expression of VDR, and enhancement of the transcription of the antibacterial cathelicidin (LL37) gene [68,69]. The exposure of monocytes to a pathogen induces 1α-hydroxylase and VDR after the pathogen is recognized by the TLR, which results in the production of cathelicidin. This cathelicidin cleaves microbial membranes and is upregulated in response to infections in humans; it acts against bacteria, viruses, and fungi [70,71]. In some patients with critical sepsis, significantly lower serum 25(OH)D and cathelicidin levels have been identified. The association between a low level of cathelicidin and death from an infectious cause has also been observed in hemodialysis patients. In addition, our previous study indicated that the presence of the C allele of −1237T/C in the TLR9 gene increases susceptibility to the development of end-stage renal disease (ESRD). Thus, patients with this functional TLR9 promoter polymorphism had a higher mean plasma IL-6 level than did those carrying the −1237TT polymorphism [72]. Excessive concentrations of IL-6 also play a role in the pathogenesis of COVID-19. Intravascular coagulation may occur, which causes multiorgan injury and endothelial dysfunction [28].

In macrophages, vitamin D suppresses NF-κB activity by upregulating the expression of IκB through the stabilization of the IκB-mRNA and a reduction in its phosphorylation. Although vitamin D has an antimicrobial effect, it also provides feedback regulation of the immune activation pathways. 1,25(OH)2D3 has been shown to potently downregulate the expression of monocyte TLR2 and TLR4, thereby suppressing the inflammatory responses that are normally activated by these receptors [73]. In the presence of 1,25(OH)2D3, dendritic cells (DCs) exhibit reduced expression of major histocompatibility complex (MHC) class II molecules and other activation markers and costimulatory makers (i.e., CD40, CD80, and CD86) [74]. This leads to reduced antigen presentation, which is accompanied by lower IL-12 secretion but increased production of tolerogenic IL-10; this then promotes the development of Th2 lymphocyte differentiation [64]. Therefore, vitamin D inhibits the maturation and differentiation of DCs, and it might be expected that treatment with vitamin D or its analogues will reduce the inflammatory response. Overall, 1,25(OH)2D3 is able to enhance the innate antibacterial defense capacity and may play a determinant role in infection in patients with AKI.

Vitamin D exerts an inhibitory action on inflammatory properties of the adaptive immune system. 1,25(OH)2D3 plays an important role in the proliferation and differentiation of T cells. Hypovitaminosis D is associated with an increased risk of autoimmune diseases, such as type 1 diabetes mellitus [75], multiple sclerosis, and inflammatory bowel disease, in humans. Suppression of the adaptive immune response could be useful for treating a variety of autoimmune diseases and for protecting transplanted organs from rejection. To date, four potential mechanisms through which vitamin D influences T-cell function have been proposed: (1) direct endocrine effects via systemic 1,25(OH)2D3; (2) direct intracrine conversion of 25(OH)D to 1,25(OH)2D3 by T cells themselves; (3) direct paracrine effects after the conversion of 25(OH)D to 1,25(OH)2D3 by local monocytes or dendritic cells; (4) an indirect effect on antigen presentation to T cells, which is mediated by localized adenomatous polyposis coli (APC) and is affected by calcitriol [76]. Vitamin D promotes a T-cell shift from Th1 to Th2, and treatment of T cells with calcitriol or analogues inhibits the secretion of the proinflammatory Th1 (IL-2, IFN-γ, and tumor necrosis factor α (TNF-α)), Th9 (IL-9), and Th22 (IL-22) cytokines [77,78] but promotes the production of other anti-inflammatory Th2 cytokines (i.e., IL-3, IL-4, IL-5, and IL-10) [79]. Active vitamin D can modulate Th2 cell responses both indirectly via the suppression of IFN-γ and IL-2 in Th1 cells and directly by influencing the expression of Th2 cytokines such as IL-4.

1,25(OH)2D3 also reduces the expression of IL-17. IL-17-producing Th17 cells play a crucial role in the induction of autoimmune disease and inflammation [80]. T cells exposed to 1,25(OH)2D3 produced significantly decreased levels of IL-17, IFN-γ, and IL-21 and had significantly increased expression of genes that are typical of regulatory T cells [81]. Regulatory T cells play an anti-inflammatory role and control autoimmune diseases by releasing IL-10 and TGF-β [82]; in addition, regulatory T cells can be induced and stimulated by 1,25(OH)2D3 through an indirect pathway, via APCs and DCs, or through a direct pathway, via an endocrine effect or the intracrine conversion of 25(OH)D to 1,25(OH)2D3 by themselves [83,84]. Thus, 1,25(OH)2D3 exerts a broad range of effects on inflammation and autoimmune diseases by reducing the number of Th17 cells and by having effects that are beneficial in terms of autoimmune and host–graft rejection; these events occur by enhancing the number of regulatory T cells. In B cells, 1,25(OH)2D3 plays an antiproliferative role that involves the inhibition of cell differentiation, the inhibition of cell proliferation, reduced initiation of apoptosis, and decreased immunoglobulin production. These effects are probably indirectly mediated by T cells [85]. Overall, 1,25(OH)2D3 is able to modulate adaptive immunity and may play a determinant role in reducing inflammation in patients with AKI. FGF23 is a protein that is synthesized by osteocytes and osteoblasts and plays a key role in the bone–parathyroid–kidney axis and in the regulation of phosphate/calcium/vitamin D metabolism [69,86,87]. FGF23 attenuates the renal production of 1,25(OH)2D3 by inhibiting the mRNA expression of CYP27B1 in the renal proximal tubule and simultaneously increasing the expression of 1,25-dihydroxyvitamin D3 24-hydroxylase (CYP24A1), which results in the generation of the inactive metabolite 24,25-dihydroxyvitamin D [88,89].

In addition to its phosphaturic effect, a recent study demonstrated that FGF23 regulates cardiomyocyte biology in a Klotho-independent manner. FGF23 was able to induce in vitro hypertrophy of cardiomyocytes, with the activation of prohypertrophic genes; this effect was dependent on the activation of the FGF receptor [90]. In patients with CKD, elevated FGF23 levels were independently associated with a greater risk of death, cardiovascular events, progression to ESRD, and premature allograft loss after kidney transplant [91,92]. Recent small studies have also reported that FGF23 increases in patients with AKI [93]. One study indicated that elevated FGF23 levels were associated with a significantly increased risk of death or the need for dialysis [94]. In 305 critically ill patients, higher urinary FGF23 levels were also independently associated with several important adverse outcomes, including greater hospital, 90 day, and 1 year mortality and longer length of stay. The study concluded that elevated FGF23 levels measured in the urine or plasma may be a promising novel biomarker of AKI, death, and other adverse outcomes in critically ill patients [95]. By using animal models, one study showed that the elevated FGF23 level was independent of PTH, vitamin D signaling, and dietary phosphate [96]. The elevated FGF23 level was consistent with patients who developed AKI after cardiac surgery and should be because of increased bone production and a longer half-life in AKI. Similarly, FGF23 can modulate peripheral immune cell function by affecting 1-alpha hydroxylase expression in monocytes and decreasing cathelicidin synthesis [3]. These data indicate that the upregulation of FGF23 may play a crucial role in defining immune responses to vitamin D, which may be a key determinant of infection in patients with AKI. The function of vitamins in innate and adaptive immunity as well as the associated process in the fight against COVID-19 are shown in Figure 2.

## 6. Vitamin D and Endothelial Dysfunction

Recent studies have found a relationship between vitamin D status and endothelial function [97]. Vitamin D therapy can improve endothelial function. In a clinical trial of patients with type 2 diabetes mellitus who had vitamin D deficiency, a one-time large dose of vitamin D improved flow-mediated vasodilation of the brachial artery and significantly decreased systolic blood pressure compared with placebo [98]. In 42 subjects with vitamin D insufficiency, normalization of 25-OH D at 6 months was associated with increases in the reactive hyperemia index and subendocardial viability ratio and a decrease in mean arterial pressure [99]. An in vitro study indicated that vitamin D may attenuate the adverse effects (including increased NF-κB expression) of advanced glycation end products on endothelial cells [100].

There is also the role of SARS-CoV-2 infection in endothelial activation and endothelial dysfunction via elevated levels of chemokines (i.e., monocyte chemoattractant protein-1), proinflammatory cytokines (i.e., interleukin-1, interleukin-6 (IL-6), and TNF-α), von Willebrand factor (vWF), and factor VIII. A review described that vitamin D maintains endothelial function by reducing the production of reactive oxygen species (ROS) as well as reducing proinflammatory mediators, such as IL-6 and TNF-α, suppressing the NF-κB pathway and attenuating lung injury by inhibiting TGF-β-induced epithelial–mesenchymal transition and stimulating type II alveolar epithelial cell proliferation and migration, reducing epithelial cell apoptosis [28]. Endothelial injury directly affects afferent arterioles and results in endothelin release and further vasoconstriction, which together cause renal microcirculatory dysfunction and induce AKI (Figure 1) [101,102].

## 7. SARS-CoV-2 and Acute Kidney Injury

The putative pathogenesis of AKI caused by COVID-19 is shown in Figure 3. SARS-CoV-2 infects both alveolar macrophages and type II alveolar cells by binding to angiotensin-converting 2 (ACE2) receptors with the receptor-binding domain (RBD) of the spike protein as a possible pathophysiological pathway. Furthermore, SARS-CoV-2 requires type 2 transmembrane protease (TMPRSS2) for the cleavage of its spike protein and to support its cell entry after binding of the RBD and ACE2. ACE2 is consumed due to the virus’ entry which, in turn, upregulates Ang II, which modulates the gene expression of several inflammatory cytokines via NF-κB signaling. In addition, infected monocytes and macrophages in the mononuclear phagocyte system also produce various proinflammatory cytokines and chemokines. Regarding the pathogenesis of AKI caused by COVID-19, intrinsic AKI has been shown to be the most common renal involvement. The process includes several pathological changes such as acute tubular injury (most common), acute interstitial nephritis, podocytopathy/collapsing focal segmental glomerulosclerosis, and thrombotic microangiopathy. Overexpression of CD147 protein also has an impact on proteinuria, and hematuria appears to be relatively prominent in COVID-19-associated AKI [103]. The mechanism of COVID-19-associated AKI includes indirect and direct causes (Figure 3). Direct viral infection of renal tubular epithelial cells, complement activation, endothelial damage, collapsing glomerulopathy and coagulopathy are probable direct causes of AKI caused by COVID-19. Indirect contributors to COVID-19-associated AKI may include organ interaction, non-COVID-19 infection, ischemic injury arising from hypotension or hypoxemia, toxic injury and possible complications of mechanical ventilation. Gastrointestinal upset and dysregulation of the Ang II pathway are other indirect contributors [103,104]. The direct or indirect actions on the kidney may also cause mitochondrial damage. Mitochondria are involved in ATP synthesis (through an efficient electron transport chain (ETC)), metabolic oxidation (via the tricarboxylic acid cycle), and full fatty acid oxidation. Mitochondria also help immune cells mature and function by reducing the generation of ROS. Persistent mitochondrial dysfunction can exacerbate AKI and increase mortality and disease mobility. Several drug targets have been reported to improve mitochondrial dysfunction including the peroxisome proliferator-activated receptor δ (PPAR δ) nuclear receptor and nicotinamide adenine dinucleotide (NAD) conservation via quinolinate phosphoribosyltransferase (QPRT) and α-amino-b-carboxy-muconate-e-semialdehyde decarboxylase (ACMSD). Another treatment option for reducing inflammation and aiding repair, such as alkaline phosphatase treatment, was also presented [105]. Melatonin has also been reported to have a number of functions in mitochondrial dysfunction, including restoring ATP generation, suppressing mitochondrial fission, preventing apoptosis in healthy cells, maintaining mitochondrial homeostasis, and improving ROS removal. In conclusion, melatonin can help mitochondria perform their regular metabolic duties while also reducing the generation of oxygen free radicals [106].

## 8. Anti-Inflammatory Effects of Vitamin D on SARS-CoV-2

Vitamin D may have the benefit of reducing the severity of COVID-19 via several pathways, such as activating monocyte (TLR1/TLR2) by pathogen-associated molecular patterns (PAMPs), enhancing antimicrobial peptide (cathelicidin and β-defensin 4A) synthesis, and increasing the generation of lysosomal degradation enzymes within macrophages. Furthermore, vitamin D has a role in adaptive immune responses to COVID-19 via endocrine, intracrine, and paracrine effects. Vitamin D not only suppresses the maturation of dendritic cells and weakens antigenic presentation but also suppresses Th1 and Th17 cytokine secretion as well as related tissue destruction. Finally, it increases cytokine production by CD4+ T cells, promotes the shift from Th1 to Th2 cells, and intensifies the efficiency of Treg lymphocytes, which results in increased humoral immunity and anti-inflammatory effects.

Furthermore, vitamin D may increase ACE2 levels to reduce the activity of the RAS by converting angiotensin I and Ang II into angiotensin 1–9 and angiotensin 1–7, respectively, which results in decreased pathophysiological effects on tissues such as inflammation and fibrosis [107]. Other benefits, such as decreasing matrix metallopeptidase 9 (MMP-9) levels and reducing bradykinin storms, have also been reported [28]. Several systematic reviews have also revealed that vitamin D supplementation is advantageous in reducing COVID-19 severity or that vitamin D deficiency is related to poor prognosis of COVID-19. It is reasonable that vitamin D may reduce the severity of COVID-19-associated AKI. In other words, vitamin D deficiency may increase the risk and severity of COVID-19-associated AKI [60,108,109].

## 9. Side Effects of Excess Vitamin D

Some studies in the past showed that either too high or too low levels of 25(OH)D could cause a poor prognosis [110,111], but subsequent studies found no definite correlation between 25(OH)D levels and outcome [112,113]. More research indicates that raising calcium levels in the blood as a result of vitamin D supplementation may be the main cause of the poor prognosis [114,115,116,117,118]. Additionally, a previous study recommended that vitamin K2 could assist in putting calcium in the hard tissues rather than the soft tissues, minimizing the likelihood of calcium-related side effects [119].

25(OH)D can exert biological activities at high concentrations by activating the VDR, and the affinity of 25(OH)D for the VDR is approximately 1000-fold less than that of 1,25(OH)2D3 [120]. Kusunoki et al. found that excess 25(OH)D exacerbates tubulointerstitial injury by modulating the kidney infiltration phenotype in mice [121]. AKI also plays a role in vitamin D toxicity. The major cause may be hypercalcemia and hyperphosphatasemia due to hypervitaminosis. The mechanism of hypercalcemia leading to AKI may include polyuria and diuresis caused by diabetes insipidus, obstruction via nephrolithiasis and renal calcification and a severe glomerular filtration rate (GFR) decrease via renal vasoconstriction. Acute phosphate nephropathy due to the tubulointerstitial deposition of phosphate calcium was mentioned as the mechanism of hyperphosphatasemia leading to AKI [16]. The safe upper limit of 25(OH)D and the benefits of vitamin D supplementation in patients with CKD and AKI still need further appropriate randomized controlled trials.

## 10. Conclusions

Vitamin D deficiency is common in COVID-19 patients and is associated with increased mortality and risk of AKI. COVID-19 can cause acute damage to the renal parenchyma through the virus directly or indirectly due to the presence of systemic factors. The kidneys of AKI patients were also more susceptible to SARS-CoV-2 infection because they had more receptors for viral entry. AKI may also cause vitamin D deficiency and increase the risk and severity of COVID-19. COVID-19 can trigger a virus-induced immune cell response, resulting in accelerated vitamin D metabolism and vitamin D consumption in the body, resulting in a decrease in vitamin D in the body. Deficiency of vitamin D activates the RAAS system in the kidneys and the whole body, which easily damages glomerular endothelial cells, podocytes, and tubular epithelial cells, thereby increasing the incidence of AKI, and it also aggravates the severity of COVID-19. The damage to the kidney tubules caused by AKI also increases FGF23 levels which, in turn, leads to lower levels of the enzyme that makes vitamin D. Furthermore, proteinuria in AKI increases the urinary loss of vitamin D. These results showed that there are cross-relationships between vitamin D deficiency, AKI, and COVID-19 (Figure 4). The efficacy and safety of vitamin D supplementation in COVID-19 patients remain controversial. Further large prospective studies evaluating the association between vitamin D and AKI in COVID-19 patients are needed before vitamin D supplementation is recommended.

## Figures and Tables

**Figure 1 ijms-23-07368-f001:**
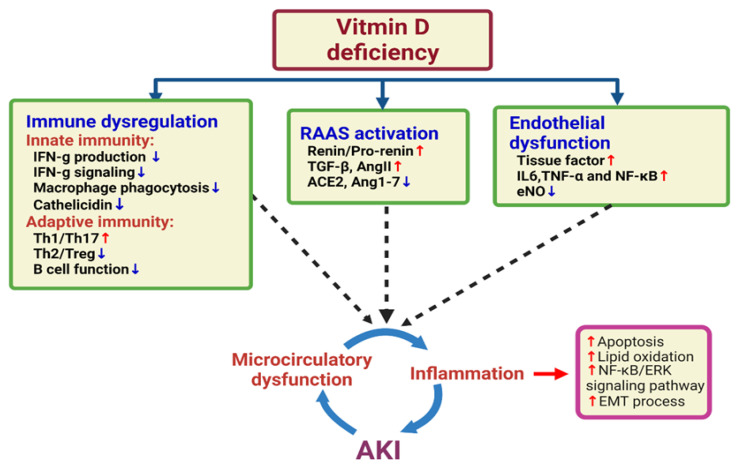
Integrated hypothesis of the association between vitamin D deficiency and acute kidney injury (AKI). Vitamin D deficiency may trigger innate and adaptive immune disorders, RAAS hyperactivity, and systemic and glomerular capillary endothelial dysfunction. All of these factors lead to direct kidney cell injury, microcirculatory dysfunction, excessive inflammation, and even macrophage activation syndrome or cytokine storms, which are key factors in the development of acute kidney injury.

**Figure 2 ijms-23-07368-f002:**
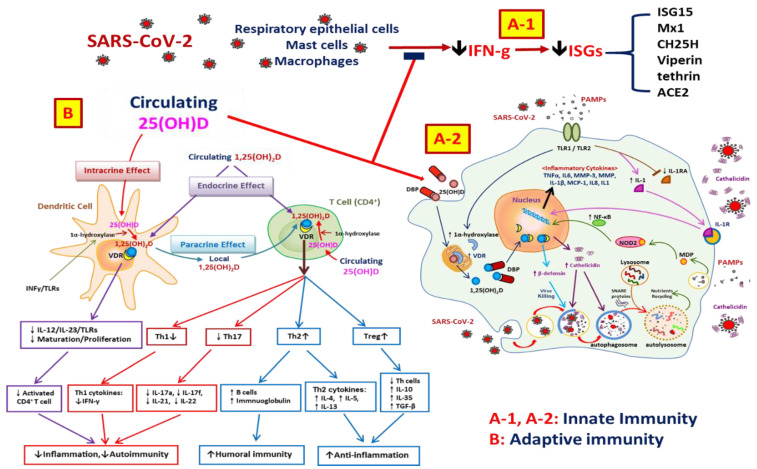
(**A**) Vitamin D-related innate immunity. SARS-CoV-2 viral proteins are able to inhibit various immune processes such as pathogen recognition, IFN production and signaling and series of interferon-stimulated genes (ISGs). Vitamin D supplement can promote IFN production and subsequent IFN signaling (**A-1**). Vitamin D binds to vitamin D receptors (VDRs) and act as a transcription factor, which induces the expression of cathelicidin and β-defensin 4A and promotes autophagy through autophagosome formation. Cathelicidin, β-defensin 4A, and mature autophagosomes then work in concert to eliminate bacteria. Vitamin D supplementation may reduce the severity of COVID-19 via enhancing the innate immune response through TLR activation and autophagy, upregulating antimicrobial peptide synthesis, and increasing the generation of lysosomal degradation enzymes within macrophages (**A-2**). (**B**) Vitamin D-related adaptive immune responses. Vitamin D can stimulate effector CD_4_^+^ cells to differentiate into one of the four types of CD_4_^+^ cells. It not only increases T helper (Th) 2 (Th2) cytokines (e.g., IL-10) and the efficiency of regulatory T (Treg) lymphocytes but also promotes the association of Th2 cells with humoral immunity. In addition, vitamin D inhibits the development of Th1 cells, which are associated with the inflammation in cellular immune response. Furthermore, vitamin D promotes the shift from Th1 to Th2 cells. Vitamin D also suppress the development of Th17 cells, which play roles in tissue damage and inflammation. Collectively, these functions may have a benefit in SARS-CoV-2 infection.

**Figure 3 ijms-23-07368-f003:**
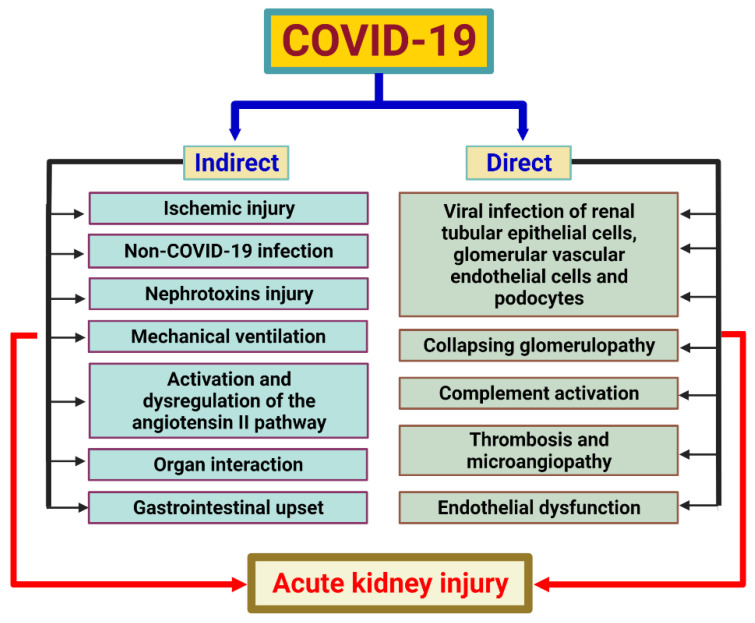
The putative pathogenesis of acute kidney injury (AKI) caused by COVID-19. The pathogenesis of AKI in patients with COVID-19 is multifactorial, which is consistent with the pathophysiology of AKI in other critically ill patients including the direct effects of SARS-CoV-2 on kidney cells and indirect effects due to the presence of systemic mechanisms. SARS-CoV-2 may exhibit viral tropism and directly affect the kidneys. Endothelial dysfunction, coagulation dysfunction, and complement activation may be important mechanisms for the development of AKI in some patients with COVID-19. The roles of systemic inflammation and immune dysfunction in the development of AKI in COVID-19 remain uncertain.

**Figure 4 ijms-23-07368-f004:**
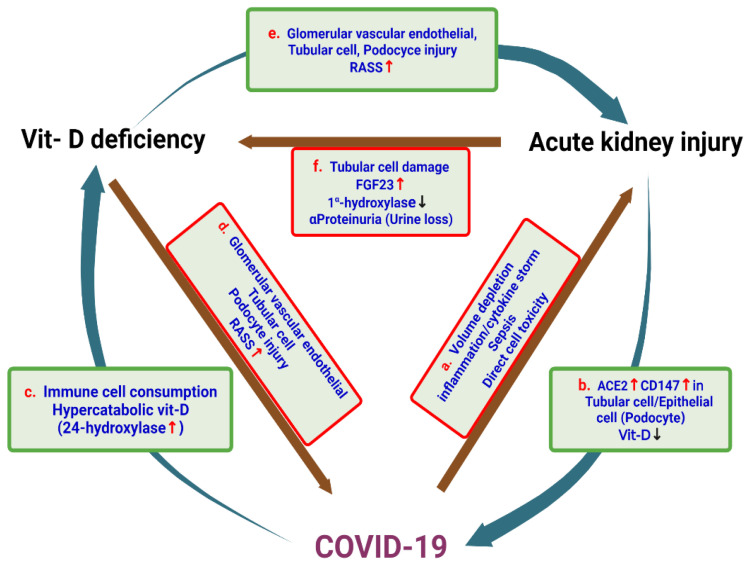
The relationships among COVID-19, AKI, and vitamin D deficiency. (**a**) COVID-19 can cause acute damage to the renal parenchyma directly by the virus or indirectly by factors such as body fluid deficiency and inflammation. (**b**) The kidneys in AKI are also more susceptible to SARS-CoV-2 infection because they have more virus entry receptors such as ACE2 and CD147. AKI can also cause vitamin D deficiency and increase the risk of COVID-19. (**c**) COVID-19 can elicit the immune cell response caused by the virus, resulting in the consumption of vitamin D; it also accelerates the metabolism of vitamin D, which leads to the decline in vitamin D in the body. (**d**) The lack of vitamin D activates the RAAS system inside the kidney and the whole body. At the same time, the lack of vitamin D can also easily cause damage to glomerular capillary endothelial cells, podocytes, and renal tubular epithelial cells, which will increase the chance of contracting COVID-19. (**e**) Vitamin D deficiency will activate the RAAS system and easily damage glomerular endothelial cells, podocytes, and renal tubular epithelial cells, thus increasing the incidence of AKI. (**f**) AKI will cause damage to the renal tubules and increase FGF23 levels, which will lead to a decrease in the concentration of enzymes that make vitamin D. Moreover, proteinuria in AKI will also increase urinary loss of vitamin D.

**Table 1 ijms-23-07368-t001:** Summary of the studies evaluating the effect of vitamin D therapy in AKI animal models.

AKI Animal Models	Intervention	Outcomes	Summary of Results
Contrast induced (Wistar albino rats) [33]	Paricalcitol i.p. for 5 days	Attenuated the increase in oxidative biomarkers; histological improvement	Antioxidant effect via the inhibition of lipid oxidation
Gentamicin induced (Sprague–Dawley rats) [34]	Paricalcitol s.c. for 14 days	Attenuated the increase in inflammatory cytokines and adhesion molecules; reversed the TGF-1-induced EMT process and extracellular matrix accumulation	Inhibition of renal inflammation and fibrosis through the interruption of the NF-κB/ERK signaling pathway, and preservation of tubular epithelial integrity via inhibition of the EMT process
Gentamicin induced (Wistar albino rats) [35]	1α,25(OH)2D3 s.c. for 8 days	Lowered blood pressure and increased urine volume by increasing GSH levels; no histological improvement	Antioxidant effect; beneficial effects via the RAS system
Ischemia/reperfusion induced (C57BL/6 mice) [36]	Paricalcitol i.p. 24 h before ischemia	Attenuated functional deterioration and histological damage; decreased Toll-like receptor 4 and nuclear translocation of the p65 subunit of NF-κB	Suppression of TLR4/NF-κB-mediated inflammation
Ischemia/reperfusion induced (Wistar albino rats) [37]	Vitamin D (0.25, 0.5, and 1 mg/kg) for 7 days before ischemia/reperfusion	Attenuated the increase in oxidative biomarkers	Activation of PPAR-γ
Cisplatin induced (Sprague–Dawley rats) [38]	Paricalcitol s.c. for 4 days	Attenuated the increase in the expression of p-ERK1/2, P-p38, fibronectin, and CTGF and proapoptotic markers CDK2, cyclin E, and PCNA	Suppression of fibrotic, apoptotic, and proliferative factors via the inhibition of TGF-β1, MAPK signaling, p53-induced apoptosis, and augmentation of p27kip1
Cyclosporin induced (Sprague–Dawley rats) [39]	Paricalcitol s.c. for 28 days	Prevented TGF-β1-induced EMT and extracellular matrix accumulation	Suppression of inflammatory, profibrotic, and apoptotic factors via the inhibition of the NF-κB, Smad, and MAPK signaling pathways
Obstructive nephropathy (CD-1 mice) [40]	Paricalcitol s.c. for 7 days	Inhibited RANTES mRNA and protein expression and abolished the ability of tubular cells to recruit lymphocytes and monocytes after TNF-β stimulation	Inhibition of renal inflammatory infiltration and RANTES expression by promoting the VDR-mediated sequestration of NF-κB signaling
Obstructive nephropathy (CD-1 mice) [41]	Paricalcitol s.c. for 7 days	Abolished TGF-β1-mediated E-cadherin suppression and α-smooth muscle actin and fibronectin induction in tubular epithelial cells by blocking the EMT directly; completely suppressed the renal induction of Snail	Preservation of tubular epithelial integrity via the suppression of the EMT
Lipopolysaccharide (LPS) induced nephropathy (CD-1 mice) [42]	Vitamin D3 (each 25 μg/kg) by gavage at 1, 24, and 48 h before LPS injection	Attenuated LPS-induced inflammatory cytokines and chemokines and adhesion molecules; reinforced the interaction between VDR and NF-κB p65 subunit in the kidney	Vitamin D3 pretreatment downregulated the renal inflammatory response, and the interaction between VDR and the NF-κB p65 subunit provided an explanation
Lipopolysaccharide (LPS) induced nephropathy (CD-1 mice) [43]	Vitamin D3 (each 25 μg/kg) by gavage at 1, 24, and 48 h before LPS injection	Alleviated LPS-induced renal GSH depletion, lipid peroxidation, serum and renal NO production, and protein nitration through regulating oxidant and antioxidant enzyme genes	Vitamin D3 pretreatment alleviated LPS-induced renal oxidative stress through regulating oxidant and antioxidant enzyme genes

## Data Availability

The data underlying this article will be shared upon reasonable request to the corresponding author.

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
