# Peer review of "The Role of Vitamin D in SARS-CoV-2 Infection and Acute Kidney Injury"

_ijms, 2022, doi:10.3390/ijms23137368_

Round 1
Reviewer 1 Report
Hsieh et al., have submitted the review entitled "The Role of Vitamin D in SARS-CoV-2 Infection and Acute Kidney Injury".
Overall the review is good. However, if the authors are ready to improve the review might gain more attention.
Here are a few suggestions to improve the manuscript.
Figure 1 requires attention; "Th1/Th17, Th2/Treg. IFN-g, oxidation" all these words should be corrected.
Figure 1 deserves more description.
Vitamin D and the immune system sections need a pictorial representation.
Line 278: CD40, CD80, and CD86 are real activation markers and co-stimulatory makers, despite their adhesive properties.
Line 363: "review revealed" ----- review described
Line 421: TLR should be abbreviated in the earlier sections.
"MMP-9" full form missing.
Author Response
The Role of Vitamin D in SARS-CoV-2 Infection and Acute Kidney Injury
Response to Reviewer 1
[General Comment]
Hsieh et al., have submitted the review entitled "The Role of Vitamin D in SARS-CoV-2 Infection and Acute Kidney Injury".
Overall the review is good. However, if the authors are ready to improve the review might gain more attention.
Here are a few suggestions to improve the manuscript
Author Reply: We sincerely appreciate your time and effort spent in reviewing this manuscript. We have revised the manuscript thoroughly according to reviewer’s suggestions. The responses to your comments are found below.
[Comment 1]
Figure 1 requires attention; "Th1/Th17, Th2/Treg. IFN-g, oxidation" all these words should be corrected.
Figure 1 deserves more description.
Author Reply:
Thank you for your valuable comments.
We have followed your suggestion to correct and describe our figure 1 as below:
Figure 1. Integrated hypothesis of the association between vitamin D deficiency and acute kidney injury (AKI). Vitamin D deficiency may trigger innate and adaptive immune disorders, RAAS hyperactivity, and systemic and glomerular capillary endothelial dysfunction. All of these factors lead to direct kidney cell injury, microcirculatory dysfunction, excessive inflammation and even macrophage activation syndrome or cytokine storms, which are key factors in the development of acute kidney injury.
[Comment 2]
Vitamin D and the immune system sections need a pictorial representation.
Author Reply:
Thank you for your valuable comments.
We have followed your suggestion to make a Figure 2 and compose some description.
Please see the Figure 2.
Figure 2. (A) Vitamin D-related innate immunity. SARS-CoV-2 viral proteins are able to inhibit various immune processes such as pathogen recognition, IFN production and signaling and series of interferon-stimulated genes (ISGs). Vitamin D supplement can promote IFN production and subsequent IFN signaling (A-1). Vitamin D bind to vitamin D receptor (VDR) and acts as a transcription factor, which induces the expression of cathelicidin and β-defensin 4A, and promotes autophagy through autophagosome formation. Cathelicidin, β-defensin 4A, and mature autophagosomes then work in concert to eliminate bacteria. Vitamin D supplementation may reduce the severity of COVID-19 via enhancing the innate immune response through TLR activation and autophagy, upregulating antimicrobial peptide synthesis, and increasing the generation of lysosomal degradation enzymes within macrophages (A-2).
(B) Vitamin D-related adaptive immune responses. Vitamin D can stimulate effector CD4+ cells to differentiate into one of the four types of CD4+ cells. It not only increases T helper (Th) 2 (Th2) cytokines (e.g., IL-10) and the efficiency of regulatory T (Treg) lymphocytes but also promotes the association of Th2 cells with humoral immunity. In addition, vitamin D inhibits the development of Th1 cells, which are associated with the inflammation in cellular immune response. Furthermore, vitamin D promotes the shift from Th1 to Th2 cells. Vitamin D also suppress the development of Th17 cells, which play roles in tissue damage and inflammation. Collectively, these functions may have benefit in SARS-CoV 2 infection.
[Comment 3]
Line 278: CD40, CD80, and CD86 are real activation markers and co-stimulatory makers, despite their adhesive properties.
Author Reply:
Thank you for your valuable comments.
We have followed your suggestion to correct. We revised the sentence as follow:
In the presence of 1,25(OH)2D3, dendritic cells (DCs) exhibit reduced expression of major histocompatibility complex (MHC) class II molecules and other activation markers and co-stimulatory makers (CD40, CD80, and CD86) [74].
[Comment 4]
Line 363: "review revealed" ----- review described
Author Reply:
Thank you for your valuable comments.
We have followed your suggestion to correct. We revised the sentence as follow:
A review described that vitamin D maintains endothelial function by reducing the production of reactive oxygen species (ROS) as well as reducing proinflammatory mediators such as IL-6 and TNF-α, suppressing the NF-κB pathway and attenuating lung injury by inhibiting TGF-β-induced epithelial mesenchymal transition and stimulating type II alveolar epithelial cell proliferation and migration, reducing epithelial cell apoptosis [28].
[Comment 5]
Line 421: TLR should be abbreviated in the earlier sections.
Author Reply:
Thank you for your valuable comments.
We have followed your suggestion to correct. We revised the sentence as follow:
such as activating monocyte TLR1/TLR2 by pathogen-associated molecular patterns (PAMPs)
[Comment 6]
"MMP-9" full form missing.
Author Reply:
Thank you for your valuable comments.
We have followed your suggestion to correct. We revised the sentence as follow:
matrix metallopeptidase 9(MMP-9)
Last, we are deeply honored by the time and effort you spent in reviewing this manuscript. In reviewing and revising our manuscript, we are motivated to read more and thus learn more from your criticisms.
Reviewer 2 Report
This manuscript is a good review of the role of vitamin D in SARS-CoV-2 infection and acute kidney injury.
The benefit of vitamin D supplements in preventing acute respiratory tract infections 102
was observed via a meta-analysis of 11,321 participants and other reviews [23].
Comment: these articles might be cited
J Endocrinol Invest 2022 Jan;45(1):167-179. doi: 10.1007/s40618-021-01639-9. Epub 2021 Jul 17. (see tables 6 and 7)
Association of Vitamin D Status and COVID-19-Related Hospitalization and Mortality. J Gen Intern Med. 2022 Jan 1:1-9. doi: 10.1007/s11606-021-07170-0.
Pre-infection 25-hydroxyvitamin D3 levels and association with severity of COVID-19 illness. PLoS One. 2022 Feb 3;17(2):e0263069. doi: 10.1371/journal.pone.0263069.
Efficacy and Safety of Vitamin D Supplementation to Prevent COVID-19 in Frontline Healthcare Workers. A Randomized Clinical Trial. Arch Med Res. 2022 Apr 18:S0188-4409(22)00045-5. doi: 10.1016/j.arcmed.2022.04.003.
The earliest evidence of the effect of vitamin D on 246
innate immunity came from the treatment of tuberculosis with cod liver oil, which is a 247
major source of vitamin D [61].
Comment: Is should be changed to was, as vitamin D supplements are now mostly from UVB-irradiated sheep’s wool lanolin.
The majority of clinical studies have found that excess and low prohormone 25(OH)D 441
levels correlate with poor prognosis, and a similar U-shaped relationship has been demon- 442
strated in cohort studies [106,107].
Comment: Which clinical studies: all health issues or those related to kidneys?
It has been suggested that U-shaped relationships regarding mortality may be due to participants with high 25(OH)D concentrations may have recently started supplementing with vitamin D, perhaps due to physicians suggesting it due to concern about osteoporosis, and that they have other underlying health issues at time of enrollment. There are more publications on this topic, listed below. Suggest discussing the topic in greater detail.
Ref. 106, Sempos, showed the increase above 100 nmol/L, which was obviously due to taking vitamin D supplements.
Ref 107, Melamed, the increase is for 25(OH)D >50 ng/mL, again only possible from vitamin D supplementation.
A Reverse J-Shaped Association Between Serum 25-Hydroxyvitamin D and Cardiovascular Disease Mortality: The CopD Study.
J Clin Endocrinol Metab. 2015 Jun;100(6):2339-46. doi: 10.1210/jc.2014-4551
A reverse J-shaped association of all-cause mortality with serum 25-hydroxyvitamin D in general practice: the CopD study.
J Clin Endocrinol Metab. 2012 Aug;97(8):2644-52. doi: 10.1210/jc.2012-1176.
Comment: In this study, “Additionally, both low and high serum levels of albumin-adjusted calcium and PTH were associated with an increased mortality risk”
Effect of calcium supplements on risk of myocardial infarction and cardiovascular events: meta-analysis
MJ Bolland, A Avenell, JA Baron, A Grey… - Bmj, 2010 - bmj.com
Calcium Supplementation, Risk of Cardiovascular Diseases, and Mortality: A Real-World Study of the Korean National Health Insurance Service Data.
Nutrients. 2022 Jun 18;14(12):2538. doi: 10.3390/nu14122538.
Compared to the control group, the hazard ratios (95% confidence intervals) of the incidence of myocardial infarction, stroke, and death in the supplementation group were 1.14 (1.03-1.27), 1.12 (1.05-1.20), and 1.40 (1.32-1.50), respectively, after adjusting for confounding variables. Considering the associated cardiovascular risk, calcium supplementation for osteoporosis treatment should be administered cautiously.
Association of coronary artery calcium with adverse cardiovascular outcomes and death in patients with chronic kidney disease: results from the KNOW-CKD.
Nephrol Dial Transplant. 2022 Jun 11:gfac194. doi: 10.1093/ndt/gfac194.
Participants with high 25-hydroxyvitamin D levels (>81 nmol/l) had lower HOMA-IR
25-Hydroxyvitamin D, insulin resistance, and kidney function in the Third National Health and Nutrition Examination Survey
M Chonchol, R Scragg - Kidney international, 2007 - Elsevier
There was no association between 25(OH)D concentrations and outcomes.
… 25-hydroxyvitamin D and 1, 25-dihydroxyvitamin D concentrations with death and progression to maintenance dialysis in patients with advanced kidney disease
J Kendrick, AK Cheung, JS Kaufman, T Greene… - … of kidney diseases, 2012 - Elsevier
Conclusion: The reverse-J shaped 25(OH)D concentration-mortality rate relationship may be due to calcium supplementation with vitamin D supplementation, rather than 25(OH)D per se.
In addition, vitamin K2 can help put calcium in the hard tissues rather than the soft tissue, thereby reducing the risk of adverse effects due to calcium:
Proper calcium use: vitamin K2 as a promoter of bone and cardiovascular health
K Maresz - Integrative Medicine: A Clinician's Journal, 2015 - ncbi.nlm.nih.gov
Author Response
The Role of Vitamin D in SARS-CoV-2 Infection and Acute Kidney Injury
Response to Reviewer 2
[General Comment]
This manuscript is a good review of the role of vitamin D in SARS-CoV-2 infection and acute kidney injury.
Author Reply: We sincerely appreciate your time and effort spent in reviewing this manuscript. We have revised the manuscript thoroughly according to reviewer’s suggestions. The responses to your comments are found below.
[Comment 1]
The benefit of vitamin D supplements in preventing acute respiratory tract infections
was observed via a meta-analysis of 11,321 participants and other reviews [23].
Tthese articles might be cited as below:
Oristrell J, Oliva JC, Casado E, Subirana I, Domínguez D, Toloba A, Balado A, Grau M. J Endocrinol Invest 2022 Jan;45(1):167-179. (see tables 6 and 7)
Seal KH, Bertenthal D, Carey E, Grunfeld C, Bikle DD, Lu CM. Association of Vitamin D Status and COVID-19-Related Hospitalization and Mortality. J Gen Intern Med. 2022 Jan 1:1-9. doi: 10.1007/s11606-021-07170-0.
Dror AA, Morozov N, Daoud A, Namir Y, Yakir O, Shachar Y, Lifshitz M, Segal E, Fisher L, Mizrachi M, Eisenbach N, Rayan D, Gruber M, Bashkin A, Kaykov E, Barhoum M, Edelstein M, Sela E. Pre-infection 25-hydroxyvitamin D3 levels and association with severity of COVID-19 illness. PLoS One. 2022 Feb 3;17(2):e0263069. doi: 10.1371/journal.pone.0263069.
Villasis-Keever MA, López-Alarcón MG, Miranda-Novales G, Zurita-Cruz JN, Barrada-Vázquez AS, González-Ibarra J, Martínez-Reyes M, Grajales-Muñiz C, Santacruz-Tinoco CE, Martínez-Miguel B, Maldonado-Hernández J, Cifuentes-González Y, Klünder-Klünder M, Garduño-Espinosa J, López-Martínez B, Parra-Ortega I. Efficacy and Safety of Vitamin D Supplementation to Prevent COVID-19 in Frontline Healthcare Workers. A Randomized Clinical Trial. Arch Med Res. 2022 Apr 18:S0188-4409(22)00045-5. doi: 10.1016/j.arcmed.2022.04.003.
Author Reply:
Thank you for your valuable comments.
We have followed your suggestion to correct. We revised the sentence and added the references as follow:
The benefit of vitamin D supplements in preventing acute respiratory tract infections was observed via a meta-analysis of 11,321 participants and other reviews [23-27].
References:
- Martineau, A.R.; Jolliffe, D.A.; Hooper, R.L.; Greenberg, L.; Aloia, J.F.; Bergman, P.; Dubnov-Raz, G.; Esposito, S.; Ganmaa, D.; Ginde, A.A.; et al. Vitamin D supplementation to prevent acute respiratory tract infections: Systematic review and meta-analysis of individual participant data. BMJ 2017, 356, i6583; DOI:10.1136/bmj.i6583.
- Oristrell J, Oliva JC, Casado E, et al. Vitamin D supplementation and COVID-19 risk: a population-based, cohort study. J Endocrinol Invest. 2022;45(1):167-179. Doi:10.1007/s40618-021-01639-9
- Seal KH, Bertenthal D, Carey E, Grunfeld C, Bikle DD, Lu CM. Association of Vitamin D Status and COVID-19-Related Hospitalization and Mortality. J Gen Intern Med. 2022 Jan 1:1-9. Doi: 10.1007/s11606-021-07170-0.
- Dror AA, Morozov N, Daoud A, Namir Y, Yakir O, Shachar Y, Lifshitz M, Segal E, Fisher L, Mizrachi M, Eisenbach N, Rayan D, Gruber M, Bashkin A, Kaykov E, Barhoum M, Edelstein M, Sela E. Pre-infection 25-hydroxyvitamin D3 levels and association with severity of COVID-19 illness. PLoS One. 2022 Feb 3;17(2):e0263069. Doi: 10.1371/journal.pone.0263069.
- Villasis-Keever MA, López-Alarcón MG, Miranda-Novales G, Zurita-Cruz JN, Barrada-Vázquez AS, González-Ibarra J, Martínez-Reyes M, Grajales-Muñiz C, Santacruz-Tinoco CE, Martínez-Miguel B, Maldonado-Hernández J, Cifuentes-González Y, Klünder-Klünder M, Garduño-Espinosa J, López-Martínez B, Parra-Ortega I. Efficacy and Safety of Vitamin D Supplementation to Prevent COVID-19 in Frontline Healthcare Workers. A Randomized Clinical Trial. Arch Med Res. 2022 Apr 18:S0188-4409(22)00045-5. doi: 10.1016/j.arcmed.2022.04.003.
[Comment 2]
The earliest evidence of the effect of vitamin D on innate immunity came from the treatment of tuberculosis with cod liver oil, which is a major source of vitamin D [61].
It should be changed to was, as vitamin D supplements are now mostly from UVB-irradiated sheep’s wool lanolin.
Author Reply:
Thank you for your valuable comments.
We have followed your suggestion to correct. We revised the sentence and added the references as follow:
The earliest evidence of the effect of vitamin D on innate immunity came from ultraviolet B (UVB)-irradiated sheep’s wool lanolin, which is a major source of vitamin D [65].
[Comment 3]
The majority of clinical studies have found that excess and low prohormone 25(OH)D
levels correlate with poor prognosis, and a similar U-shaped relationship has been demonstrated in cohort studies [106,107].
Which clinical studies: all health issues or those related to kidneys?It has been suggested that U-shaped relationships regarding mortality may be due to participants with high 25(OH)D concentrations may have recently started supplementing with vitamin D, perhaps due to physicians suggesting it due to concern about osteoporosis, and that they have other underlying health issues at time of enrollment. There are more publications on this topic, listed below. Suggest discussing the topic in greater detail.The reverse-J shaped 25(OH)D concentration-mortality rate relationship may be due to calcium supplementation with vitamin D supplementation, rather than 25(OH)D per se.In addition, vitamin K2 can help put calcium in the hard tissues rather than the soft tissue, thereby reducing the risk of adverse effects due to calcium
Author Reply:
Thank you for your valuable comments.
We have followed your suggestion to correct. We revised the sentence and added the references as follow:
Some studies in the past showed that either too high or too low levels of 25(OH)D could cause a poor prognosis [110,111], but subsequent studies found no definite correlation between 25(OH)D levels and outcome [112,113]. More research indicates that raising calcium levels in the blood as a result of vitamin D supplementation may be the main cause of the poor prognosis [114-118]. Additionally, a previous study recommended that vitamin K2 could assist in putting calcium in the hard tissues rather than the soft tissues, minimizing the likelihood of calcium-related side effects [119].
References:
- Sempos, C.T.; Durazo-Arvizu, R.A.; Dawson-Hughes, B.; Yetley, E.A.; Looker, A.C.; Schleicher, R.L.; Cao, G.; Burt, V.; Kramer, H.; Bailey, R.L.; et al. Is there a reverse J-shaped association between 25-hydroxyvitamin D and all-cause mortality? Results from the U.S. nationally representative NHANES. J. Clin. Endocrinol. Metab. 2013, 98, 3001–3009; DOI:10.1210/jc.2013-1333.
- Melamed, M.L.; Michos, E.D.; Post, W.; Astor, B. 25-hydroxyvitamin D levels and the risk of mortality in the general population. Arch. Intern. Med. 2008, 168, 1629–1637; DOI:10.1001/archinte.168.15.1629.
- Kendrick J, Cheung AK, Kaufman JS, et al. Associations of plasma 25-hydroxyvitamin D and 1,25-dihydroxyvitamin D concentrations with death and progression to maintenance dialysis in patients with advanced kidney disease. Am J Kidney Dis. 2012;60(4):567-575. DOI:10.1053/j.ajkd.2012.04.014
- 113. Chonchol M, Scragg R. 25-Hydroxyvitamin D, insulin resistance, and kidney function in the Third National Health and Nutrition Examination Survey. Kidney Int. 2007;71(2):134-139. DOI:1038/sj.ki.5002002
- 114. Bolland MJ, Avenell A, Baron JA, et al. Effect of calcium supplements on risk of myocardial infarction and cardiovascular events: meta-analysis. BMJ. 2010;341:c3691. Published 2010 Jul 29. DOI:10.1136/bmj.c3691
- 115. Durup D, Jørgensen HL, Christensen J, Schwarz P, Heegaard AM, Lind B. A reverse J-shaped association of all-cause mortality with serum 25-hydroxyvitamin D in general practice: the CopD study. J Clin Endocrinol Metab. 2012;97(8):2644-2652. DOI:10.1210/jc.2012-1176
- 116. Durup D, Jørgensen HL, Christensen J, et al. A Reverse J-Shaped Association Between Serum 25-Hydroxyvitamin D and Cardiovascular Disease Mortality: The CopD Study. J Clin Endocrinol Metab. 2015;100(6):2339-2346. DOI:1210/jc.2014-4551
- 117. Park JM, Lee B, Kim YS, et al. Calcium Supplementation, Risk of Cardiovascular Diseases, and Mortality: A Real-World Study of the Korean National Health Insurance Service Data. Nutrients. 2022;14(12):2538. Published 2022 Jun 18. DOI:10.3390/nu14122538
- 118. Jung CY, Yun HR, Park JT, et al. Association of coronary artery calcium with adverse cardiovascular outcomes and death in patients with chronic kidney disease: results from the KNOW-CKD [published online ahead of print, 2022 Jun 11]. Nephrol Dial Transplant. 2022;gfac194. DOI:10.1093/ndt/gfac194
- 119. Maresz K. Proper Calcium Use: Vitamin K2 as a Promoter of Bone and Cardiovascular Health. Integr Med (Encinitas). 2015;14(1):34-39.
Last, we are deeply honored by the time and effort you spent in reviewing this manuscript. In reviewing and revising our manuscript, we are motivated to read more and thus learn more from your criticisms.